# In Vitro Culture Studies for the Mitigation of Heavy Metal Stress in Plants

**DOI:** 10.3390/plants12193387

**Published:** 2023-09-25

**Authors:** Doaa Elazab, Maurizio Lambardi, Maurizio Capuana

**Affiliations:** 1IBE—Institute of BioEconomy, National Research Council (CNR), 50019 Florence, Italy; doaa.elkassas@ibe.cnr.it (D.E.); maurizio.lambardi@ibe.cnr.it (M.L.); 2Department of Pomology, Faculty of Agriculture, Assiut University, Assiut 71526, Egypt; 3IBBR—Institute of Biosciences and Bioresources, National Research Council (CNR), 50019 Florence, Italy

**Keywords:** alleviation, metals, metalloids, plant growth regulators, rhizobacteria

## Abstract

Heavy metals are among the most common and dangerous contaminants; their action on plants, as well as the possibility for plants to effectively absorb and translocate them, have been studied for several years, mainly for exploitation in phytoremediation, an environmentally friendly and potentially effective technology proposed and studied for the recovery of contaminated soils and waters. In this work, the analysis has focused on the studies developed using in vitro techniques on the possibilities of mitigating, in plants, the stress due to the presence of heavy metals and/or improving their absorption. These objectives can be pursued with the use of different substances and organisms, which have been examined in detail. The following are therefore presented in this review: an analysis of the role of metals and metalloids; the use of several plant growth regulators, with their mechanisms of action in different physiological phases of the plant; the activity of bacteria and fungi; and the role of other effective compounds, such as ascorbic acid and glutathione.

## 1. Introduction

The accumulation of heavy metals in soils and waters is a global, ever-increasing problem, particularly for agricultural soils, considering the possible transfer of toxic elements into the human food chain [1,2]. Among the various methods of remediation of polluted environments, phytoremediation is probably the most promising, basing its action on the use of plants to extract, sequester, and/or detoxify pollutants [3].

The first reports on the possible use of plants for soil and water decontamination date back to the 1980s, and since then many publications on this subject have appeared (see, e.g., reviews [2,4]). In the last decades, several approaches have been developed, and many herbaceous (with some hyperaccumulators) and woody (mainly fast-growing) species have been indicated [5,6,7] also highlighting the possible different behavior of different genotypes (clones, cultivars) within a species [8].

The vast existing literature has mainly investigated how different species react to the presence of heavy metals in the soil; however, now a new aspect of the topic is gaining interest, which can be summarized as the set of actions that lead plants to mitigate the stress due to the presence of heavy metals and/or improve their absorption. This review focused on the use of in vitro culture for the study of these actions. It was deemed appropriate to better circumscribe an otherwise too vast topic, also considering that in vitro cultures present some peculiar aspects worthy of particular attention.

Tissue culture and in vitro propagation can be considered useful tools for understanding the processes of pollutant uptake by plants and all the mechanisms involved [9], and, in general, for the assessment of tolerance to environmental stresses since stress conditions can be easily circumscribed and controlled in vitro [10]. The complex and variable nature of the different abiotic stresses that can interact under field or greenhouse conditions can make it difficult to study the response of plants to these factors; furthermore, there are constraints on the assessment of tolerance to abiotic stresses under field conditions that may be extremely variable [11]. The use of methods based on in vitro tissue culture, therefore, can be an important tool for a better understanding of the physiology and biochemistry of plants cultured under stress conditions [12,13,14]. The in vitro model systems using cell and organ culture allow for easier sample handling and data retrieval than experiments under field conditions [15,16]. Furthermore, the differences in environmental conditions that we find in vivo, such as variations and interactions among hundreds of substances in the soil, the influence of different simultaneous stresses, interplay among tissues, and competition caused by interorganismic interactions, are drastically reduced in in vitro culture systems due to their defined nutrient media, lighting conditions, temperature, humidity, and homogeneous stressor application. It should also be noted that some substances for the mitigation of the effects of heavy metals on plants have only been tested in vitro, as reported later in this review. On the other hand, it should be taken into account that some aspects of in vitro culture may limit the extensibility of the results obtained with this system. This is because the particular cultivation conditions are a primary source of stress for the plants, which are raised in conditions of nutrition, lighting, humidity, and temperature different from the natural ones. The results of in vitro experiments conducted on this topic are of extreme importance to better understand the mechanisms of stress mitigation due to heavy metals. However, it should also be underlined that these findings require confirmation or modulation based on tests conducted in nature on plants in the field. Balancing the advantages and disadvantages of in vitro culture, however, the large existing literature demonstrates the importance of this technique for carrying out investigations on the aforementioned topic, and therefore it was deemed appropriate to offer a broad overview.

This review offers an overview of studies carried out with in vitro culture techniques on the main substances and organisms used for the mitigation of heavy metal stress in plants and, in some cases, for the consequent better metal uptake. In each paragraph, a brief introduction is presented on the main characteristics and mechanisms of the actions of these substances, with bibliographical references not necessarily linked to the in vitro environment. Subsequently, the main results obtained through in vitro culture research are illustrated.

## 2. Metals and Metalloids

It is well known that some metals, such as iron (Fe), manganese (Mn), copper (Cu), and zinc (Zn), are essential micronutrients for plants, required in trace amounts as they act as cofactors for various enzymes. These and others—e.g., cobalt (Co), cadmium (Cd), nickel (Ni), selenium (Se), mercury (Hg), and the metalloids arsenic (As) and silicon (Si)—have long been accumulated in soils through human activities such as industrial wastes, fertilizer applications, smelting, and sewage disposal [17] and can become toxic to plants and humans [18].

It has been documented in many studies that the supply of different metals, especially micronutrients, can influence the uptake and distribution of some heavy metals and eventually reduce their accumulation in plants. However, probably the main activity of metals towards the mitigation of stress caused by heavy metals occurs through competitive action [19,20].

In vitro experiments involving different metals are illustrated below.

### 2.1. Zinc

Zn is a vital micronutrient required for plant growth [21]. It can significantly increase tolerance towards HMs by increasing the accumulation of potential osmolytes and reducing the production of reactive oxygen species (ROS) [22]. Zn also affects the uptake of HMs and their translocation [23] and has been reported to modulate the toxicity effects of several heavy metals, particularly Cd, in a wide range of plant species [24,25]. Zn and Cd, in fact, share the same chemical, physical, and geochemical properties [26]. The alleviation of Zn against Cd is mainly based on a competition mechanism that produces a reduction of the metal uptake from the soil [27] and/or the sequestration of Cd in the roots [28]. Elazab et al. [29] provided experimental evidence of the remarkable role of Zn with two concentrations (200 and 500 mM) in the mitigation of Cd (500 mM) stress in banana shoots in vitro, finding that Zn supplementation can increase both total flavonoid production and ascorbic acid levels in plants exposed to Cd stress. Furthermore, they observed that Zn supplementation inhibits the accumulation of proline, an amino acid that tends to increase under stress conditions. Chakravarty and Srivastava [30] reported that the presence of Zn^2+^ at equimolar concentrations with Cd^2+^ (0.1 and 0.2 mM) can affect the uptake and translocation patterns of both elements in flax (*Linum usitatissimum* L.) plantlets in vitro. This interaction could potentially reduce cadmium toxicity by modifying its uptake and distribution within the plant. They found that, when both zinc and cadmium were present in equal amounts, the adverse effects of Cd toxicity were mitigated. One of the observed effects was that the regenerated plantlets showed higher Cd uptake in their roots than in their shoots. On the other hand, Zn was translocated from roots to shoots, overcoming the movement of Cd. Wajda et al. [31] found that Zn ions at (4.84 µg/mL) counteracted the negative effect of Cd (from 5.62 to 22.48 µg/mL) on the growth of *Catharanthus roseus* callus, observing that zinc reduced the accumulation and translocation of Cd from roots to shoots. These findings are in agreement with Sánchez-Thomas et al. [32] which demonstrated that chronic exposure to high Zn^2+^ (less than 0.5 mM) promoted protection against Cd^2+^ toxicity in *Euglena gracilis* cells, with zinc involved in glutathione (GSH) and polyphosphate ligand activity. Ma et al. [33] demonstrated that 30 µM of Zn alleviated the toxicity of Cd (30 µM) in the presence of Si in rice *(Oryza sativa*) cell walls by binding Zn^2+^ to ligands through the formation of the Zn-[Si-hemicellulose matrix] complexes, which limit the uptake of Cd. Recently, the application of metal and metal-oxide nanoparticles (NPs) in agriculture has started to arouse wide research interest for their efficiency to mitigate abiotic environmental stressors and pollutants. For instance, Karmous et al. [34] evaluated the potential of zinc oxide nanoparticles (ZnONPs) at different concentrations (2.5, 5, and 10 mg L^−1^) in monitoring Cd toxicity on in vitro-cultured pepper plants, stating that the application of ZnONPs could recover 70–85% of shoot length, 40–50% of shoot fresh weight, 60–80% of root length, and 70% of root fresh weight.

### 2.2. Silicon

Si is not considered an essential element for most higher plants [35]; however, there is growing evidence of its beneficial effects on plant growth and development in a wide variety of plant species [36,37]. Numerous studies have revealed the ability of Si to restore various biotic and abiotic stresses [38,39]. Si can provide protection and alleviate the harmful effects of heavy metals by strengthening the cell wall. The deposition of silicon in cell walls can fortify them, providing a physical barrier against the entry of heavy metals into plant cells. This deposition helps maintain the integrity and functionality of the cell wall, preventing its degradation and subsequent cell death [40,41]. The other mechanism of Si for relieving HM stress is to stabilize various cellular organelles, including mitochondria, Golgi bodies, the endoplasmic reticulum (ER), and the nucleus. By protecting these organelles from metal-induced damage, Si helps maintain their structure and function, thereby promoting cell survival. These results were confirmed by Prabagar et al. [42] who used a cell suspension of Norway spruce (*Picea abies*) to investigate the interactions between Si and alluminium (Al). They observed that the presence of Si resulted in a reduction of free Al concentration in the cell wall, proposing that the formation of aluminosilicate complexes was a potential mechanism for this reaction. Their results showed that the addition of 1.0 mM Si reduced the toxicity of Al in Norway spruce cells. They also demonstrated that changes in cell suspension culture (such as cell wall thickening, degree of vacuolization, and the degeneration of cellular organelles such as mitochondria, Golgi bodies, ER, and nucleus) are associated with the process of cell death in plants under Al stress. However, the presence of silicon was found to have a significant ameliorative effect on these cellular changes. Moreover, Si can limit toxic mineral uptake through Si deposition in roots as well as the chelation or complexation of toxic metals [43]. Si has a significant alleviating effect on Al toxicity in rice seedling cells in vitro and in vivo [44,45]. Rossini et al. [46] stated that the presence of Si can mitigate the Cd stress on *Alternanthera tenella* plants, thanks to an enhancement of photosynthetic pigments and a consequent greater accumulation of biomass. The effect of Si on the attenuation of Cd-induced stress was demonstrated on poplar callus [47]. Si has the same ameliorative effect on poplar callus exposed to As [48]. Li et al. [49] investigated the role of Si in improving the tolerance of *Poa annua* shoots in vitro to Cd stress and demonstrated that Si restored the growth of Cd stress-inhibited shoots. They also observed that Si significantly reduced the uptake of Cd in the roots; therefore, the percentage of Cd in the roots is markedly reduced under Cd + Si compared to those treated with Cd alone. Liu et al. [50] described the results of cell and protoplast cultures of rice (*Oryza sativa*) in the presence of a combination of Cd and Si. They found that, when Si was added at a concentration of 1.0 mM, there was a significant accumulation of Si in the cell walls. Furthermore, the study observed that the addition of Si resulted in a significant reduction in Cd levels within the protoplasts compared to the Si-untreated ones. This reduction was more pronounced at high Cd concentrations (up to 60 mM). Ma et al. [51] confirmed these results and stated that cells accumulating Si significantly reduce the net Cd^2+^ influx compared to cells not treated with Si. Nanotechnology offers a further approach to metal remediation through the design and use of micron- or nano-size metal-binding polymer particles with very high specific surface area [52] and high reactivity [53]. In this context, Cui et al. [54] studied the impact of silica nanoparticles (SiNPs) on the attenuation of Cd toxicity in rice using a cell suspension culture and found that the presence of SiNPs allowed cells to remain almost intact even with high concentrations of Cd. Silicic acid, a neutral compound that moves through membrane channels with water, is the form of silicon that plants can uptake and accumulate [55]. Mono-silicic acid (H_4_SiO_4_) has the unique ability to bind with heavy metal ions, thus protecting plants from excess HM [56]. The toxic effect of Al can be attenuated thanks to the formation of Al-Si complexes, which reduce the free toxic form of Al in water [57]. Corrales et al. [58] stated that *Zea mays* L. plants pretreated with 1 mM silicic acid for 72 h, then exposed to Al in a nutrient solution, showed lower Al uptake and longer roots than non-pretreated plants. They suggest that the attenuating effect of silicic acid is due to the exclusion of Al from the root tips in Si-pretreated plants. Quintal-Tun et al. [59] tested an Al-sensitive cell suspension line of *Coffea arabica* L. to study the effect of Al on the phosphoinositide signal transduction pathway. The results indicated that Al has a major influence on phosphatidic acid, which produces lipids essential for plant cell membrane formation and regulates developmental processes responsible for attenuating environmental stresses. Silicic acid plays an essential role in protecting the formation of PA against the action of Al and in maintaining the stability of cell membranes.

### 2.3. Selenium

The use of Se, thanks to its biochemical functions, represents one of the most effective strategies to mitigate the toxic effects of metals. If applied at a low concentration, it is useful in regulating plant development by alleviating the damage induced by environmental stress [60]. In particular, Se supplementation can limit metal uptake by roots and translocation to shoots, which is one of the vital mechanisms for metal/metalloid stress tolerance [61]. Furthermore, Se reduces oxidative stress in plants against metal stresses [62]. Such roles of low-dose Se in plant growth as antioxidants, stress relievers, and inhibitors of the absorption of heavy metals, including Cd [63], chrome (Cr) [64], lead (Pb) [65], and Cu [66] have been reported.

In vitro, the effect of Se on Cd-treated chloroplasts was examined by measuring the physicochemical properties of their membranes. Chloroplasts were isolated from *Brassica napus* seedlings cultured on media containing Cd and/or Se, and the results suggested that Se may partly counterbalance the negative effects of Cd. This protective action led to an increase in the size of chloroplasts and a partial reconstitution of their ultrastructure [67]. Rice cell suspensions were treated with Cd in the presence or absence of Se. Se pretreatment was found to alleviate Cd toxicity by increasing the percentage of live cells by 83.1%. Furthermore, Se increased the lignin content and cell wall thickness, thereby increasing the mechanical strength of the cell walls, as determined by atomic force microscopy [68].

### 2.4. Other Metals

The study conducted by López-Serrano et al. [69] examined the impact of Fosetyl-Al, a systemic fungicide commonly used for the control of grapevine diseases caused by oomycetes, on grapevine cell suspension cultures. They suggested that the inhibitory effect of Al on the peroxidase enzyme could be overcome by the presence of Ca^2+^ and Mg^2+^ ions. They also proposed the “apoplastic Ca^2+^/Mg^2+^ displacement hypothesis” to explain the mechanism of action of Al on peroxidase in grapevine suspension cells. In plant cells, a significant portion of the total calcium (Ca) content is found in the apoplast, i.e., the space outside the plasma membrane but within the cell wall. Apoplastic Ca^2+^/Mg^2+^ plays a crucial role in maintaining plasma membrane stability and providing cross-links between negatively charged cell wall polymers, thus contributing to cell cohesion.

Recent studies have shown that Ca, an alkaline earth metal, could be used as an exogenous substance to protect plants against Cd stress by attenuating growth inhibition, regulating metal uptake and translocation, enhancing photosynthesis, mitigating oxidative damage, and improving the control of signal transduction in plants. Ca may alleviate Cd toxicity in tobacco and corn [70,71]. The role of magnesium (Mg) in alleviating HMs toxicity was studied by Rengel and Robinson [72] who observed that Al^3+^ and Mg^2+^ are similar; therefore, the Mg^2+^ absorption system and metal-binding sites in enzymatic reactions may not distinguish well between Al^3+^ and Mg^2+^ ions. Competition between Al^3+^ and Mg^2+^ ions have been demonstrated for membrane transporters and metal binding sites in enzymatic reactions [73].

Iron (Fe) is a key element in plants, playing an important role as a cofactor in several metabolic processes. Moreover, it is crucial in heme- and iron-sulfur proteins [74], which are essential for maintaining chloroplast structure and the performance of photosynthesis [75]. It was found that Fe competes with Cd for the same pathways in plant cells [76], thus Fe can be absorbed instead of Cd, minimizing Cd uptake and improving the physiological functions of plants [77]. Elazab et al. [29] investigated the effect of two concentrations of Fe (200 and 500 μM) on the toxic effect of Cd on banana shoots in vitro. The results demonstrated that the addition of Fe at both concentrations was able to counteract the negative effects of 500 μM Cd, which was evidenced by the increase of photosynthetic pigments, non-enzymatic antioxidants, total protein, and non-protein thiol content in Fe-treated versus untreated plantlets.

Magnesium oxide (MgO) is an essential functional metal oxide with a high surface area that acts as an adsorbent and has been used for the remediation of toxic metals [78]. Hussain et al. [79] studied the effect of MgO on seed germination, growth, biomass, total phenolics and flavonoids, and the antioxidant potential of in vitro-growing radish (*Raphanus sativus* L.) under Pb stress. They found that MgO nanoparticles in combination with thidiazuron can play a significant role in the production of secondary metabolites and Pb phytoaccumulation, as well as improving seed germination and growth.

## 3. Plant Growth Regulators

Plant growth regulators (PGRs) are a varied category of organic compounds that are produced naturally at low concentrations in plant tissues. They either improve or prevent plant growth and control the morphological, biochemical, and physiological aspects of plant development [80]. PGRs are active compounds that are very important in inducing a stress response in plants [81]. Abiotic stresses cause many changes in the level of endogenous phytohormones, leading to growth inhibition with the aim of reducing lesions. A first reaction to an abiotic stress is the reduction in the concentration of auxins, gibberellins, and cytokinins, combined with an increase in the content of abscissic acid (ABA), jasmonates, and salicylic acid [82,83]. Other regulatory components, such as polyamines, peptides, and organic acids, also play a role in the response of plants to stress, acting as immunomodulators [84]. It is argued that exogenous application of PGRs can improve plant survival under HM stress, preventing biomass decrease and enhancing photosynthesis; moreover, in the presence of HMs, PGRs stimulate antioxidant activity [85]. Several physiological processes are regulated by endogenous phytohormones, such as auxins and cytokinins, which are responsible for the integration of growth control and stress response [86]. Consequently, the concentrations of these phytohormones could be influenced by stress circumstances and also by external hormonal treatment. However, information on the effect of exogenous PGRs application on the natural phytohormone profile of plants under HMs toxicity stress is still incomplete. Moreover, it has been reported in previous studies that PGRs can be used to enhance the accumulation of HMs in plants, playing an efficient role in alleviating the toxicity of HMs [87].

As concerns in vitro culture, Hussain et al. [79] investigated the effects of different concentrations of thidiazuron (TDZ) and/or MgO NPs on germination, growth, and biomass of *Raphanus sativus* L. and on the alleviation of Pb toxicity. They confirmed that TDZ, alone or with MgO NPs, significantly improves seed germination and plays an important role in secondary metabolite production and Pb phytoaccumulation (up to 50 mg L^−1^) in radish explants.

2,4-Dichlorophenoxyacetic acid (2,4-D) has been tested for HMs toxicity alleviation. 2,4-D is a synthetic compound with auxin-like action that significantly interferes with important physiological processes in plants [88]. Having a carboxylate group in its structure, 2,4-D can react with metal ions such as Co^2+^ and Ni^2+^ forming complexes sparingly soluble in water, leading to increased mobility and a consequent lower HMs accumulation in plants [89].

Indole-3-butyric acid (IBA) is one of the synthetic auxins generally used in tissue culture and plant propagation due to its higher stability and efficiency compared to IAA. It has been reported in many studies that IBA can alleviate the inhibitory effects of HM toxicity on plant growth and development and promote adventitious rooting even under HM stress by regulating enzymatic and non-enzymatic ROS scavenging systems and the activity of IAA-oxidase [90]. Li et al. [91] observed that IBA treatment on mung bean [*Vigna radiata* (L.) Wilczek] can mitigate the Cd repressive effects on in vitro adventitious rooting. They showed that IBA counteracted the impact of Cd toxicity on rooting by regulating the antioxidative plant system and, specifically, the IAA-oxidase activity.

Gibberellic acid (GA_3_) has been reported to have a role in the regulation of cell membrane permeability and in the control of membrane transport processes, causing a reduction in metal uptake [92]. In this context, Wiszniewska et al. [93] demonstrated that exogenous application of 10 μM GA_3_ enhanced the reaction of in vitro *Daphne jasminea* shoots to nickel (Ni) stress, with stimulation of growth rate and proliferation. Ni tolerance in the presence of GA_3_ has been attributed to peroxisomal reactions that increase the synthesis of endogenous jasmonic acid.

Other substances are included in the category of growth regulators, namely ABA, salicylic acid (SA), and jasmonic acid (JA). When added exogenously, they can increase tolerance to abiotic stresses [94]. These acids interfere with cell metabolism and can stimulate biological defense to alleviate the toxic effects of HMs [94]. HMs are intensely phytotoxic due to the production of ROS and the formation of malondialdehyde (MDA) [95], which damage cell membranes, proteins, lipids, nucleic acids, and photosynthetic pigments such as chlorophyll and carotenoids [96]. Plants have evolved antioxidant mechanisms to inhibit the oxidative stress and lipid peroxidation caused by HMs [97]. The enzymatic antioxidant system includes glutathione reductase (GR), superoxide dismutase (SOD), peroxidase (POD), ascorbate peroxidase (APX), catalase (CAT), and non-enzymatic antioxidants (glutathione, ascorbate), which continuously scavenge harmful ROS, although under conditions of severe stress, antioxidants may not have sufficient capacity to reduce the toxic effect of HMs [98]. Hence, it is appropriate to add signal molecules that mediate stress tolerance for a better understanding of how plants respond to HM stress.

SA, an important signaling molecule, is responsible for eliciting certain reactions in response to biotic and abiotic stresses. SA has been reported to be essential for the regulation of plant growth, development, and responses to environmental stresses [99]. Several studies have stated that SA helps to inhibit the uptake and/or accumulation of HMs in plant organs and increases the accumulation of nutrients (e.g., K, Ca, Mg, Mn, and Fe) [100,101,102]. SA application generally improves the anti-stress response of cells against oxidative damage, including the regulation of genes related to ROS metabolism [103]. Furthermore, SA may have a role in protecting the stability and integrity of cell membranes [104], and in enhancing the performance of the photosynthetic machinery against HM toxicity [103]. In a study to evaluate the effect of Cd on two successive subcultures of banana shoots in vitro, it was observed that a treatment with SA plays an important role in improving the response to Cd toxicity, even at high concentrations of the metal [105].

As a primary defense signaling hormone, JA regulates plant growth and enhances the plant defense system against HMs toxicity and various biotic [106] and abiotic [107] stresses by stimulating the signaling pathways, resulting in increased plant resistance to stressful conditions. JA has been associated with limiting HMs accumulation and improving tolerance to toxic elements by managing the ion transport system, chelating capacity, and antioxidant enzyme activity in plants [108]. JA can stimulate signaling pathways, thus resulting in increased plant resistance to abiotic stress conditions. Furthermore, JA has shown a protective effect on the plant against many HMs, such as Cu and Cd, and in *Arabidopsis thaliana*, a positive response with the improvement of photosynthesis, vegetative growth, and chlorophyll and carotenoid content [109]. Piotrowska et al. [110] tested the effect of exogenously applied JA on the growth and metabolism of *Wolffia arrhiza* (Lemnaceae) cultured in vitro under Pb stress. JA at 0.1 μM showed a positive effect, helping plants avoid cumulative damage following Pb exposure and significantly alleviating its negative effects on shoots, such as growth inhibition, biomass reduction, and chlorosis. JA improved plant growth, increased the fresh weight of shoots, photosynthesis (chlorophyll a and carotenoids), and protein content. JA-induced metal stress tolerance is closely related to the obstruction of Pb entry into cells and the activation of the antioxidant defense system to reduce metal-induced oxidative damage. These data are in line with [93], who found that exogenous application of 0.5 μM JA improved the multiplication and all growth parameters of *Daphne jasminea* shoots under Ni exposure in vitro. They indicated that the uptake of Ni was decreased with JA treatment due to limited transpiration and stomatal closure.

ABA is a plant stress hormone that is rapidly produced to relieve various plant stresses [111]. It is fundamental in plant responses to a range of abiotic stresses, such as excess light, salinity, nutrient deficiency, and heavy metals [112,113,114]. It plays many important roles in the plant defense system, mainly by promoting stress tolerance and activating genes involved in the control of antioxidant defense systems [115]. The physiological effects of ABA in plants under HM have been repeatedly demonstrated. For example, ABA enhanced the activities of APX and CAT and reduced the activity of SOD, thereby alleviating HM-induced oxidative stress and toxicity [116]. It has been argued that endogenous ABA is strongly correlated with the Cd tolerance of rice seedlings and could exert its regulatory effect on transpiration rate, resulting in a reduction in Cd translocation to shoots [117]. Some studies have demonstrated the role of ABA in the regulation of the antioxidant defense in HM tolerance, in particular the formation of adventitious roots under HM stress. Li et al. [90] examined the effects of ABA, as a pre-treatment, on antioxidative enzymes and antioxidants through adventitious root formation in mung bean [*Vigna radiata* (L.) Wilczek] in vitro under Cd stress and confirmed that ABA pre-treatment neutralized Cd stress by promoting the activity of antioxidant enzymes, e.g., APX and POD, and elevated SOD and CAT activities and ascorbic acid (ASA) levels, which were decreased due to Cd stress, thus alleviating the toxic effect of Cd.

## 4. Microorganisms

Microorganisms can influence plants' responses to abiotic stresses, including metal stress, through different mechanisms [118]. The ability of microorganisms to bind and/or convert heavy metals present in soil solutions and free water is a developing system to eliminate these contaminants and reduce their harmful effects [119]; therefore, the use of bacteria and fungi is the basis of bioremediation techniques as ecological alternatives to physical-chemical treatments to remove or reduce pollutants [120].

Toxic HMs are converted by bacteria or their enzymes into less harmful poisonous forms during bioremediation, which aids in the cleanup of polluted areas [121]. In different bacterial plasmids and chromosomes, studies have demonstrated the role of specific genes against various HM toxicities [122]. *Arthrobacter*, *Micrococcus*, *Pseudomonas*, and *Bacillus* have all been identified as promising candidates as biocontrol agents for use in metal adsorption technologies due to their ability to immobilize metals such as Cd, Cu, Cr, and Pb [123,124]. Furthermore, bacteria can improve nutrient uptake, stimulating growth and defenses and reducing the intake of heavy metals and their harmful consequences [125].

In vitro studies by Salomon et al. [126] indicated that *Bacillus licheniformis* and *Pseudomonas fluorescens* play a role as antistress by inducing ABA synthesis when they colonize the roots of *Vitis vinifera*, cv. Malbec, grown in vitro. *Rhizobacteria* have several strategies to promote plant growth, including the production of siderophores; these organs have a strong affinity for ferric ions, thus increasing iron uptake by plants and activating plant defense mechanisms against biotic and abiotic stresses, including HMs. [127]. The toxicity of arsenite in grapevine in vitro culture was detected by Pinter et al. [128]; they reported that bacterization with plant growth-promoting rhizobacteria (PGPR) could improve plant growth and reduce the toxic effects of arsenite in grapevine, cv. Malbec, cultivated in vitro. This study was conducted to evaluate the effects of different bacterial strains on the growth and tolerance of grapevine plantlets in vitro in the presence of NaAsO_2_ (sodium arsenite). *Bacillus licheniformis*, *Micrococcus luteus*, and *Pseudomonas fluorescens* were selected based on their As(III) tolerance and plant growth-promoting (PGP) traits. In vitro-grown grapevine plantlets were inoculated with selected PGP strains and treated with or without As. In the presence of As, all strains increased the activity of catalase, which helps to break down hydrogen peroxide. An improvement in both plant biomass and protein content was observed with *M. luteus*, while with *B. licheniformis*, there was only an increase in plant biomass. Both strains are therefore considered promising candidates for bioremediation of As(III) contamination. On the contrary, *P. fluorescens* proved to be less tolerant to As(III) and showed less significant effects on plant growth.

PGP bacterial isolates of the family Bacillaceae (*Bacillus* and *Halobacillus*) were tested on *Arachis hypogaea* cultured in vitro in the presence of Zn, Al, and Pb under salinity (1% NaCl) [129]. The isolates demonstrated a positive impact on different plant physiological parameters (lesser lignification, intact protoxylem, and cortical parenchyma), suggesting their usefulness in the alleviation of HM plant stress. The activity of potential zinc-tolerant bacteria belonging to the genus *Serratia* for improving in vitro growth of *Zea mais* plants under Zn toxicity was explored by Jain et al. [130]. Four strains identified as *Serratia* sp. showed high tolerance to Zn and exhibited various plant growth-promoting activities, such as growth improvement, enhancement of antioxidant enzyme activities, and decreased Zn accumulation. Moreover, the strains produced gluconic acid as a natural chelating agent for heavy metals.

In an in vitro system, endophytic bacteria from *Murdannia spectabilis*, belonging to the genera *Bacillus*, *Pantoea*, *Microbacterium*, *Curtobacterium*, *Chryseobacterium*, *Cupriavidus*, *Siphonobacter*, and *Pseudomonas*, were tested with and without Zn and Cd stress [131]. Treatment with Zn plus Cd was shown to support the persistence of *Cupriavidus plantarum* MDR5, and inoculation did not weaken the plants. Plant growth and Zn/Cd accumulation were not significantly affected by Zn-/Cd-tolerant endophytes, and consequently, *Cupriavidus plantarum* MDR5 was proposed for application in bioaugmentation processes.

A study was focused on rhizobacteria living with adapted plants (*Stipa tenuissima*, *Sulla spinosissima*, and *Acacia cyanophylla*) in heavy metal-polluted mining soils [132]. In this work, the isolated bacteria (*Rhodococcus qingshengii* and *Pseudarthrobacter oxydans* strains) passed through in vitro pre-screening tests concerning heavy metal tolerance, and, before performing planta tests by means of microbial inoculation, the characterization of those bacterial strains still uncharacterized was completed in vitro.

Probiotics such as *Lactobacillus* spp. have displayed the ability to bioremediate heavy metals under in vitro conditions. Goyal et al. [133] demonstrated that *Bacillus clausii*, a probiotic species of the genus *Bacillus*, showed a strong ability to survive and tolerate high concentrations of Cr, Cd, Pb, and Ni, thus appearing to be a good candidate for HM alleviation in vivo. In an in vitro co-culture of *Arabidopsis arenosa* and the basidiomycete *Sporobolomyces ruberrimus*, Jędrzejczyk et al. [134] found that plants inoculated with the fungus showed significantly fewer stress symptoms in medium containing excess Fe, Zn, and Cd. It was also observed that the fungus was able to precipitate Fe in the medium, thus limiting the exposure of plants to metal toxicity.

Arbuscular mycorrhizal fungi (AMF) are crucial microorganisms of the rhizosphere that establish symbiotic associations with a broad range of terrestrial plants. These associations improve resistance to biotic and abiotic stresses, such as heavy metal toxicity. The mechanisms by which AMFs provide stress tolerance involve promoting nutrient absorption and enhancing the activities of antioxidant enzymes [135]. For in vitro tolerance against HMs toxicity, Blaudez et al. [136] examined 31 ectomycorrhizal isolates of *Paxillus involutus*, *Pisolithus tinctorius*, *Suillus bovinus*, *S. luteus*, and *S. variegatus* on Cd, Cu, Ni, and Zn-enriched media. The tolerance was measured by the inhibition of biomass production, and the data showed that *S. luteus*, *S. variegatus*, and *P. tinctorius* have a higher tolerance to Cd, Cu, and Zn, in contrast to *P. involutus*, which is more tolerant to Ni. This variation in metal tolerance can be attributed to several factors, including genetic differences, physiological adaptations, and the evolutionary history of fungi. These results are examined in light of the possible advantages of ectomycorrhizal fungi in protecting their host plants from metal toxicity. Khade and Adholeya [137] studied the responses in vitro of carrot hairy roots obtained by transformation via root-inducing transferred DNA of *Agrobacterium rhizogenes* (Ri-TDNA), and AMF established these roots in Pb-amended media. They found that the dual cultures of transformed roots and AMF (*Glomus lamellosum*, *G. intraradices*, and *G. proliferum*) exhibited tolerance to 5 ppm of Pb, also observing that, when subjected to Pb stress, in the mycorrhized transformed roots of carrot, the total phenol content increased in the roots and exudation into the medium decreased. 

AMF have repeatedly been demonstrated to alleviate heavy metal stress in plants. Hildebrandt et al. [125] presented data on extraradical mycelia (ERM) of *Glomus intraradices* cultured in vitro with different heavy metals (Cd, Cu, or Zn), showing that the expression of genes encoding proteins potentially involved in heavy metal tolerance varied in their response to different heavy metals. Glomalin is an insoluble glycoprotein produced on the hyphae of AMF. In an in vitro experiment, González-Chávez et al. [138] demonstrated its possible mitigating effect founding that glomalin from hyphae of an isolate of *Gigaspora rosea* sequestered up to 28 mg Cu/g.

Within this contest, it can be mentioned that in vitro root culture and hairy-root culture are both interesting systems for achieving metal removal under controlled conditions, as reviewed by Santos-Díaz [139] about species such as *Rubia tinctorum*, *Scirpus americanus*, and *Typha latifolia* treated with Cd, Cu, Ni, and Zn.

## 5. Other Compounds

### 5.1. Ascorbic Acid

ASA is one of the most important non-enzymatic antioxidants and plays a vital role in the growth and normal functioning of plants. It has high solubility in water and can directly and indirectly decrease ROS. ASA regulates many of the cellular processes, such as cell division, cell differentiation, and senescence [140]. Similarly, ASA protects lipids and proteins and improves tolerance against various abiotic stresses, further promoting plant growth, transpiration, photosynthesis, and oxidative defense potential; furthermore, it functions as a cofactor for enzyme activity and modulates signaling pathways and physiological processes via phytohormones [141].

The application of ASA can improve the tolerance of plants to abiotic stresses, thus improving the level of photosynthesis and vegetative growth. In fact, it acts as a powerful antioxidant against the action of heavy metals by directly eliminating ROS, regenerating other antioxidants, chelating heavy metals, and inducing detoxification enzymes. These mechanisms contribute to the protective effects of ASA acid against heavy metal-induced oxidative stress and cellular damage. [141]. In vitro culture of *Lycium barbarum* L. under Pb stress on medium enriched with 1 mM ASA resulted in better apical growth and improved biochemical parameters, such as proline, MDA, chlorophylls, and carotenoid production [142]. These results were also confirmed by Saleem et al. [143], who stated that in three genotypes of *Saccharum officinarum* L. cultured in vitro at different concentrations of Pb, a pre-treatment with ASA led to better callus growth and regeneration than in untreated explants; improved antioxidant activity of POD, SOD, and CAT was also found.

### 5.2. Nitric Oxide

Certain plant regulators or signaling molecules, such as nitric oxide (NO), have the ability to manage abiotic stresses. NO is considered a biological messenger that essentially acts in the regulation of physiological processes in plants, such as plant growth and reaction against biotic and abiotic stresses [144]. The commonly used donor of NO is sodium nitroprusside (SNP). Generally, NO is produced endogenously in plants via nitrate reductase and NO synthase and participates in abiotic stress response mechanisms, including salinity, drought, heat, and HM stress [145]. NO can eliminate ROS produced as a result of the toxic effects of HMs on plants, regulating the enzymatic and non-enzymatic antioxidant systems to increase the tolerance to HMs stress in plants. Furthermore, NO can modulate related-gene expression and related protein activity as essential strategies to alleviate HM toxicity [146].

Several studies have reported the effects of exogenous application of SNP on the attenuation of HM toxicity in plants. Xiong et al. [147] reported that exogenous application of NO counteracts the toxic effect of Cd by regulating the cellular distribution of excess Cd and its accumulation in the cell walls of rice plants. Khairy et al. [148] investigated the role of NO in relieving Cd and Cu stress on in vitro-grown tobacco (*Nicotiana tabacum*), observing the effect of the metals on plant growth, chlorophyll content, and activity of rubisco and rubisco activase, which are important proteins involved in atmospheric carbon fixation. Their results showed an improvement in growth and total chlorophyll content in the presence of SNP under Cd/Cu stress. Furthermore, supplementation with SNP showed higher contents and activities of rubisco and rubisco activase compared to control and Cu/Cd-stressed plants.

### 5.3. Glutathione

Glutathione (GSH, γ-glutamyl-cysteinyl-glycine) is a widely distributed tripeptide found at millimolar concentrations in plant cells [149]. It is produced from three amino acids in two ATP-dependent steps, starting with the formation of a peptide bond between γ-glutamate and cysteine by γ-glutamylcysteine synthetase (GSH1), followed by the addition of glycine catalyzed by glutathione synthetase (GSH2) [150]. GSH is a substrate for phytochelatin synthesis and is essential for the detoxification of heavy metals such as Cd and Ni [151]. GSH is involved in numerous cellular processes, particularly in the defense against ROS [152], and can also block heavy metals by forming non-toxic complexes with metals and facilitating their sequestration away from sensitive sites in cells [151].

Numerous studies have revealed that plant exposure to high levels of HM induces ROS. GSH contributes to the regulation of H_2_O_2_ levels in plant cells [152] GSH reduction acts as an antioxidant and is directly involved in the reduction of most ROS generated during HM stress [152]. Furthermore, GSH acts as a fundamental factor in many cellular detoxification processes due to heavy metals. GSH performs this role after activation and association with these metals [153]. The coupling between GSH and HMs is regulated by glutathione S-transferase [154], with subsequent transfer to the vacuole, where it performs the action of protecting the cell from the toxic effect of HM [155,156]

Shankar et al. [157] evaluated the role of exogenous GSH application in improving the resistance against stress of different HMs. The study evaluated the effect of adding different concentrations (1, 5, 10, or 25 mg L^−1^) of GSH to media containing four different HMs [As_2_O_3_, CuSO_4_, ZnSO_4_, or Pb(NO_3_)_2_] on the in vitro growth of *Spilanthes calva* L. seedlings. GSH at 10 mg L^−1^ was optimal for maximum contrast to the negative effects of all HMs on morphogenesis. The addition of GSH to Pb-containing medium resulted in a significant improvement in almost all vegetative growth parameters. The effect on the morphogenic response of GSH on *Mucuna pruriens* L. plantlets growing in vitro on HM-containing media was investigated by Alam et al. [158] They used MS medium supplemented with 5.0 μM CuSO_4_ or 80.0 μM ZnSO_4_, adding GSH in order to assess the level of specific glutathione-S-transferase (GST) activity. The results showed that GST was effectively involved in relieving heavy metal stress.

## 6. Conclusions

This article proposes an overview of possible interventions to mitigate the stress due to the presence of heavy metals or to improve their absorption. The conspicuous literature analyzed in this review, summarized in Table 1, testifies to the validity of in vitro cultures as a tool for scientific investigations. Although it is appropriate to take due consideration of some limitations related to in vitro techniques, there is no doubt that, thanks to these, it has been possible to make important steps forward in the knowledge of the relationship between plants and heavy metals and on possible interventions to limit their toxicity. A dutiful premise is that the vast subject of genetic transformation, for which in vitro culture techniques represent an essential tool, has not been taken into consideration unless such techniques have been used for the consequent studies on stress mitigation. The main strategies provided by the aforementioned alleviation techniques include the action of other heavy metals, which mainly involves a sort of competitive activity; the use of growth regulators, which play important roles in various physiological mechanisms of the plant; the contribution of bacteria, which is part of the big topic of bioremediation; and the role of other substances that have demonstrated efficacy in this context. It is undeniable that there is still much to be studied on all these topics and that the in vitro culture tool can provide further valid support. The final objective, in addition to increasing basic knowledge on the biochemical and physiological aspects of the plant-metal relationship, is to achieve a practical and advantageous application of the mitigation techniques.

## Figures and Tables

**Table 1 plants-12-03387-t001:** List of major agents analyzed in in vitro culture experiments for their actions in mitigating the effect of heavy metal stress on plants and improving metal uptake.

Mitigating Agent	Crop	Heavy Metal	Tolerance Strategy Induced by the Mitigating Agent	Reference
Zn	*Musa* spp.	Cd	shares the same chemical and physical properties with Cd; it competes with Cd and reduces its uptake by plants	[29]
Zn	*Linum usitatissimum* L.	Cd		[30]
Zn	*Catharanthus roseus*	Cd		[31]
Zn	*Euglena gracilis*	Cd		[32]
Zn	*Oryza sativa*	Cd		[33]
ZnONPs	*Capsicum annuum* L.	Cd		[34]
Si	*Picea abies*	Al	strengthens the cell wall, providing a physical barrier against the entry of heavy metals into plant cells; binds, in some cases, with heavy metals ions	[42]
Si	*Alternanthera tenella*	Cd		[46]
Si	*Populus* spp.	Cd		[47]
Si	*Poa annua*	Cd		[49]
Si	*Oryza sativa*	Cd		[50]
Si	*Populus* spp.	As		[48]
SiNPs	*Oryza sativa*	Cd		[54]
Silicic acid	*Coffea arabica* L.	Al		[59]
Se	*Brassica napus* L.	Cd	increases lignin content and cell wall thickness	[67]
Se	*Oryza sativa*	Cd		[68]
Mg	*Vitis vinifera* L.	Al	Mg ions are similar to Al ions; therefore, plants take up Mg instead of Al	[69]
Ca	*Vitis vinifera* L.	Al	Ca ions regulate metal uptake and translocation in plants	[69]
Fe	*Musa* spp.	Cd	Fe competes with Cd for the same pathways in plant cells	[29]
MgO	*Raphanus sativus* L.	Pb	acts as an absorbent	[79]
TDZ	*Raphanus sativus* L.	Pb	improves the activity of the root system, with an increase in the absorption of water and nutrients	[79]
IBA	*Vigna radiata* (L.) Wilczek	Cd	promotes adventitious rooting under HM stress	[91]
GA3	*Daphne jasminea*	Ni	increases cell membrane permeability	[93]
SA	*Musa spp.*	Cd	enhances antioxidant defenses, inhibiting HM uptake	[105]
JA	*Wolffia arrhiza*	Pb	stimulates signaling pathways resulting in increased plant resistance	[110]
JA	*Daphne jasminea*	Ni		[93]
ABA	*Vigna radiata* L. Wilczek	Cd	activates the genes of the antioxidant defense systems	[90]
Rhizobacteria	*Vitis vinifera* L. cv. Malbec	As	increase catalase activity leading to hydrogen peroxide decomposition and an increase in biomass	[128]
*Bacillus* and *Halobacillus*	*Arachis hypogaea*	Zn, Al, Pb	increase catalase activity and enhances plant biomass	[129]
*Serratia* spp.	*Zea mais*	Zn	enhance the antioxidant enzymes activities and decreases Zn accumulation	[130]
*Sporobolomyces ruberrimus*	*Arabidopsis arenosa*	Fe, Zn, Cd	precipitates Fe in the medium, causing lower exposure of plants to metal toxicity	[134]
AMF	Carrot hairy roots	Pb	promote nutrient absorption and enhance the activities of antioxidant enzymes	[137]
AMF	*Glomus intraradices*	Cd, Cu, Zn		[125]
ASA	*Lycium barbarum* L.	Pb	promotes antioxidants activity, increases fresh and dry weight of plant	[142]
ASA	*Saccharum officinarum* L.	Pb		[143]
NO	*Nicotiana tabacum*	Cd	regulates the enzymatic and non-enzymatic antioxidant system	[148]
GSH	*Spilanthes calva* L.	As, Cu, Zn, Pb	regulates H_2_O_2_ level in plant cells; fundamental factor in cellular detoxification processes, also binding to HMs	[157]
GSH	*Mucuna pruriens* L.	Cu, Zn		[158]

## Data Availability

Not applicable.

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
