# Peer review of "In Vitro Culture Studies for the Mitigation of Heavy Metal Stress in Plants"

_plants, 2023, doi:10.3390/plants12193387_

Round 1

Reviewer 1 Report

1. This review article is very challenging to read, not only due to the poor writing skills but also their unclear aims.

2. Abstract: It should be rewritten to improve the grammar and define more clearly about their main theme. According to the manuscript title, the authors seem to focus on phytoremediation but actually, they describe more about the impacts of nutritional metals in normal in vitro cultivation of plants. In my opinion, the authors should pay their most attention to in vitro heavy-metal stress conditions and summarize the impact of phytoremediation.

3. All subtitles in this manuscript need to be improved to make the readers understand more clearly the main theme of each subtopic.

4. The authors should give more paragraphs to comprehensively describe the molecular mechanisms of phytoremediation and the achievements when using in vitro systems to investigate. It's better to organize 1-2 conclusive graphs to present the mechanisms.

5. Table 1: Very confusing data, don't understand what it means. It should be focused on stress and remediation but not the cases about normal concentrations of nutritional metals.

6. The authors should collect some photos of treated plants, cellular events, and molecular evidence on in vitro phytoremediation.

Author Response

REVIEWER #1:

  1. This review article is very challenging to read, not only due to the poor writing skills but also their unclear aims.   R: We were surprised by this comment, because the manuscript was checked by a native English-speaking scientist and  also because none of the other reviewers expressed the same opinion. Reviewer #1, whose comments we respect and appreciate however, indicated that he is not an expert in the English language. Regarding the “unclear aims”, we have made changes to the Introduction to improve understanding, as suggested.

  1. Abstract: It should be rewritten to improve the grammar and define more clearly about their main theme. According to the manuscript title, the authors seem to focus on phytoremediation but actually, they describe more about the impacts of nutritional metals in normal in vitro cultivation of plants. In my opinion, the authors should pay their most attention to in vitro heavy-metal stress conditions and summarize the impact of phytoremediation.   R: Thank you for the comment, we have changed the title and abstract, to be more consistent with the content.

  1. All subtitles in this manuscript need to be improved to make the readers understand more clearly the main theme of each subtopic.   R: Thank you for your comment. However, we do not understand what wrong as for the subtitles. A clearer explanation or some examples from the Reviewer would have helped us understand. Therefore, we have decided to keep the subtitles as they are.

  1. The authors should give more paragraphs to comprehensively describe the molecular mechanisms of phytoremediation and the achievements when using in vitro systems to investigate. It's better to organize 1-2 conclusive graphs to present the mechanisms.   R: Thank you for the comment. Indeed, the term “phytoremediation” was not entirely appropriate for the review, and may cause confusion for the reader. We have therefore changed the title, making it more responsive to the purposes of the review. So, it is not necessary to explain the molecular mechanism.

  1. Table 1: Very confusing data, don't understand what it means. It should be focused on stress and remediation but not the cases about normal concentrations of nutritional metals.   R: Sorry, we do not understand what is unclear. The caption clearly states “List of major agents analyzed in in vitro culture experiments for their actions in mitigating the effect of heavy metal stress on plants and improving metal uptake”. So, Table 1 is referring to the mitigating action of each specific agent, used in experimental trials (and not in “normal concentrations”) on the cited heavy metal. However, we slightly changed the Table header to make this concept even clearer.

  1. The authors should collect some photos of treated plants, cellular events, and molecular evidence on in vitro phytoremediation.   R: A common strategy in the compilation of reviews is not to include photos for two main reasons: (i) not to show preferences, favoring the work of a few authors compared to the total number of works cited; (ii) not having to ask permission from the authors for any photos included. We have therefore respected this review compilation strategy, followed in the vast majority of published reviews.

Reviewer 2 Report

Review of the manuscript « In vitro cultures and phytoremediation: studies for the mitigation of heavy metal stress in plants

The paper reviews the present knowledge on the action of added compounds and microorganisms to plant in vitro cultures with the aim to better know the mechanisms of mitigating the cell reactions to metal pollution.

 -        This topic is important, and the review gives a good overview for the reader.

-        The paper is within the journal’s topic of heavy metal stress but not necessarily of “phytomonitoring and phytoremediation”.

-        It can be accepted by PLANTS, however, it needs some specifications and clarifications, some final touch.

Title and abstract:

-        Applications of in vitro experiments with heavy metals are mainly restricted to cultures in a watery environment. I suggest removing the “phytoremediation” in title and abstract, a term which is most often used for plants growing in the field. In the text it can be used in specific cases.

-        Possibilities of mitigation metal stress in entire plants are not reviewed, but the mechanisms in plant tissues and cells are studied by adding of different substances and organisms to in-vitro cultures.

Introduction:

-        Remove author names when citing by numbers.

-        Not only cultivation conditions of nutrition, lighting, humidity and temperature are different from the natural ones but also:

o   the latter conditions vary in the field in time and space,

o   variations and interactions among hundreds of substances occur in the soil,

o   interactions among different simultaneous stresses are present,

o   interplay among tissues is important,

o   allocation and translocation mechanisms are specific,

o   interorganismic interactions include competition…...

-          It is missing in the introduction that the extensibility of the results obtained by cell and tissue cultures or special test organisms such as Arabidopsis is still far from practical application with some exceptions to be named in the review.

Nevertheless, such reviews are important for further research.

Metals and metalloids:

-        It is nickel, not nichel.

-        Results on Co, Mn, Ni, and Hg are missing (?). It must be stated that papers studying these heavy metals using in vitro cultures are missing - or why they were not included?

-        “However, probably the main activity of metals towards the mitigation of stress caused by heavy metals occurs through a competitive action [18-19]”. This cannot stand alone. Metal uptake and allocation is strongly influenced and mitigated by different root tissues. It is first the ectomycorrhizal, epidermal, and endodermal barrier in roots and the safe allocation and complexation of metals in less physiological active tissues and cell compartments (which is also part of the later text in the review). Such tissue aspects are only missing when cells are directly exposed to the metals in cell cultures.

In general, for all paragraphs:

-        It is often (and it should be consistently) the first paragraph, which refers to entire plant or plant organ reactions (please specify). It is recommended to distinguish entire plant reactions from tissue cultures (which tissue?) and from cell cultures in a follow-up very clearly in each paragraph.

-        Furthermore, the culture must be more specifically reported (hydroculture, in agar, in soil etc.) because it influences the result.

-        Particularly in Table 1 a column of the culture properties is missing.

-        It should be checked that it is always precisely written, what “reduced toxicity”, or “protection from toxicity”, or “mitigation of toxicity” etc. means (chemical process, growth promotion, complexation, trapping or protective layer…), because it differs in different sentences.

-        Correct the different fonts in the microorganism paragraph.  

-        Correct underscores in reference 88 and 107, correct certain fonts, line spacing, blanks and tabs in the reference list.  

The paragraphs with Zn addition:

-        The Zn concentrations (and other metal concentrations, Cd) are missing. Their influence is decisive.

-        Zn influence on Cd is repeatedly reported (can be shortened).

The following paragraphs:

-        Are there papers, which indicate, that pants grown in soils rich in Si, Se, or Mg are more tolerant to heavy metal contamination?

Plant growth regulators:

-        Why do the “other compounds” not belong to the “growth regulators” rather than be a new chapter after the microorganisms?

-        2,4-D: Why is an increased ion mobility responsible for lower HM accumulation?

Author Response

REVIEWER #2:

The paper reviews the present knowledge on the action of added compounds and microorganisms to plant in vitro cultures with the aim to better know the mechanisms of mitigating the cell reactions to metal pollution.

 -        This topic is important, and the review gives a good overview for the reader.   R: We sincerely thank you for the comments.

-        The paper is within the journal’s topic of heavy metal stress but not necessarily of “phytomonitoring and phytoremediation”.   R: Yes, we agree, thank you.

-        It can be accepted by PLANTS, however, it needs some specifications and clarifications, some final touch.   R: Thank you.

Title and abstract:

-        Applications of in vitro experiments with heavy metals are mainly restricted to cultures in a watery environment. I suggest removing the “phytoremediation” in title and abstract, a term which is most often used for plants growing in the field. In the text it can be used in specific cases.

R: Thank you for the suggestion. We have changed the title, deleting the term “phytoremediation”, and focused attention on the mitigation of stress due to heavy metals. The term “phytoremediation” has been left only in specific appropriate cases.

-        Possibilities of mitigation metal stress in entire plants are not reviewed, but the mechanisms in plant tissues and cells are studied by adding of different substances and organisms to in-vitro cultures.

R: Thank you for your appropriate comment. 

Introduction:

-        Remove author names when citing by numbers.

R: Yes, it has been done.

-        Not only cultivation conditions of nutrition, lighting, humidity and temperature are different from the natural ones but also:

o   the latter conditions vary in the field in time and space,

o   variations and interactions among hundreds of substances occur in the soil,

o   interactions among different simultaneous stresses are present,

o   interplay among tissues is important,

o   allocation and translocation mechanisms are specific,

o   interorganismic interactions include competition…...

R: We thank for the appropriate comment. The point has been modified accordingly (“such as variations…”).

-          It is missing in the introduction that the extensibility of the results obtained by cell and tissue cultures or special test organisms such as Arabidopsis is still far from practical application with some exceptions to be named in the review.

R: We agree. A sentence on this point has been inserted (“The results on in vitro experiments…”).

Nevertheless, such reviews are important for further research.

R: Thank you.

Metals and metalloids:

-        It is nickel, not nichel.

R: Corrected in the text.

-        Results on Co, Mn, Ni, and Hg are missing (?). It must be stated that papers studying these heavy metals using in vitro cultures are missing - or why they were not included?

R: Thank you so much. There were not any former in vitro studies on the mitigation of the toxic effects of Co, Mn, Ni and Hg.

-    “However, probably the main activity of metals towards the mitigation of stress caused by heavy metals occurs through a competitive action [18-19]”. This cannot stand alone. Metal uptake and allocation is strongly influenced and mitigated by different root tissues. It is first the ectomycorrhizal, epidermal, and endodermal barrier in roots and the safe allocation and complexation of metals in less physiological active tissues and cell compartments (which is also part of the later text in the review). Such tissue aspects are only missing when cells are directly exposed to the metals in cell cultures.

R: The quoted sentence does not refer to the toxic action of heavy metals, but only to the mitigating effects of other metals on HM stress.

In general, for all paragraphs:

-        It is often (and it should be consistently) the first paragraph, which refers to entire plant or plant organ reactions (please specify). It is recommended to distinguish entire plant reactions from tissue cultures (which tissue?) and from cell cultures in a follow-up very clearly in each paragraph.

R: As typical of in vitro culture, all the studies, with very few exceptions, are based on shoot cultures. So, we didn’t consider necessary to specify this concept for every citation.

-        Furthermore, the culture must be more specifically reported (hydroculture, in agar, in soil etc.) because it influences the result.

R: All cited articles on HM stress mitigation concern in vitro culture, as indicated in the title and mentioned in the introduction.

-        Particularly in Table 1 a column of the culture properties is missing.

R: As stated above, we only reviewed papers on in vitro culture.

-        It should be checked that it is always precisely written, what “reduced toxicity”, or “protection from toxicity”, or “mitigation of toxicity” etc. means (chemical process, growth promotion, complexation, trapping or protective layer…), because it differs in different sentences.

R. Thank you. Indeed, the mode of action of each substance used has been mentioned in detail in the manuscript. Some of them are mitigating toxicity by influencing the absorption of HMs, e.g. Zn, Fe and Si, and some of them are increasing tolerance to HMs, e.g. SA and JA. Using words like “alleviate, enhance and improve… etc.” still indicates an action on the protection of HM toxicity.

-        Correct the different fonts in the microorganism paragraph.

R: Done.

-        Correct underscores in reference 88 and 107, correct certain fonts, line spacing, blanks and tabs in the reference list. 

R: Done.

The paragraphs with Zn addition:

-        The Zn concentrations (and other metal concentrations, Cd) are missing. Their influence is decisive.

R: Thank you, we added the concentrations.

-        Zn influence on Cd is repeatedly reported (can be shortened).

R: Sorry, it is reported only when necessary. We do not understand where “to shorten”.

The following paragraphs:

-        Are there papers, which indicate, that plants grown in soils rich in Si, Se, or Mg are more tolerant to heavy metal contamination?

R: Si, Se and Mg are added to the soils as fertilizers, and there are papers on their roles in influencing the growth and the tolerance to heavy metals. A specific review could be done on this topic, but it is beyond the scope of this review.

Plant growth regulators:

-        Why do the “other compounds” not belong to the “growth regulators” rather than be a new chapter after the microorganisms?

R: Ascorbic acid, glutathione and nitric oxide do not belong to the PGR category, so we have classified them into different sections so as not to confuse them with other compounds.

-        2,4-D: Why is an increased ion mobility responsible for lower HM accumulation?

R: Sorry. It was a mistake. It is a greater mobility of the soluble complex which leads to less absorption by the plant.

‘ion’ deleted.

Reviewer 3 Report

I don’t have much to say. The manuscript "In vitro cultures and phytoremediation: studies for the mitigation of heavy metal stress in plants" is a good summary of the topic based on extensive literature, including the latest. It is advisable for the authors to indicate precisely in conclusions what gaps in our knowledge related to the topic need to be filled.

Below are my detailed comments:

Page 2

Nichel(Ni) – please correct

Page 3

These findings are in agreement with [31] which … - better: ” These findings are in agreement with study results by Sánchez-Thomas et al. [31]”

… less than 0.5 mM - please use SI units only (mmol); check this issue in the whole paper

…presence of Zn+2 … concentration with Cd+2… -  this notation refers to the oxidation state…you should use the notation for ions (Zn2+ and Cd2+); please check this issue in the whole paper

Page 6

up to 50 mg/l - better: mg/dm3

Page 7

metal ions like Co+2 and Ni+2 - rather Co2+ and Ni2+

Page 9

to 5 ppm of Pb - please use SI units (mg/dm3)

Page 10

atmospheric carbon fixation - url ???

Page 11

(1, 5, 10, or 25 mg/l); 10 mg/l - mg/dm3

Page 13

It is undeniable that there is still much to be studied on this matter… - please provide more details

References

Almost all the references require correction according to the journal’s requirements

Author Response

REVIEWER #3:

I don’t have much to say. The manuscript "In vitro cultures and phytoremediation: studies for the mitigation of heavy metal stress in plants" is a good summary of the topic based on extensive literature, including the latest. It is advisable for the authors to indicate precisely in conclusions what gaps in our knowledge related to the topic need to be filled.

R: Thank you for the good comment to the review.

Below are my detailed comments:

Page 2

Nichel(Ni) – please correct

R: Done in the whole text.

Page 3

These findings are in agreement with [31] which … - better: ” These findings are in agreement with study results by Sánchez-Thomas et al. [31]”

R: Thank you, we considered this notice on the whole text.

… less than 0.5 mM - please use SI units only (mmol); check this issue in the whole paper

R: ‘mM’ is the most largely way to indicate millimolar concentrations. Therefore, we prefer to leave this indication.

…presence of Zn+2 … concentration with Cd+2… -  this notation refers to the oxidation state…you should use the notation for ions (Zn2+ and Cd2+); please check this issue in the whole paper

R: Notations corrected

Page 6

up to 50 mg/l - better: mg/dm3

R: We preferred to change as ‘mg L-1’ that, together with mg/l and mg/L, is much more common in scientific papers than mg/dm3.

Page 7

metal ions like Co+2 and Ni+2 - rather Co2+ and Ni2+

R: Done. It is on page 6

Page 9

to 5 ppm of Pb - please use SI units (mg/dm3)

R: Please see the comment above

Page 10

atmospheric carbon fixation - url???

R: Thank you for the notice, it was misprinting and has been corrected.

Page 11

(1, 5, 10, or 25 mg/l); 10 mg/l - mg/dm3

R: Please see the comment above

Page 13

It is undeniable that there is still much to be studied on this matter… - please provide more details

R: The sentence has been modified. We also believe that all the fields of research mentioned deserve further investigation, so a generic conclusion seems appropriate to us, also to avoid repetitions and lengthening of the text.

References

Almost all the references require correction according to the journal’s requirements

R: Done.

Reviewer 4 Report

The review is quite well written and organized. 

Author Response

REVIEWR #4:

The review is quite well written and organized.

R: Thank you very much for your comment.

Round 2

Reviewer 1 Report

I don't have further questions.